# LET'S LET'S LET'S LET'S...
# UNDERSTAND LOOPING IN REASONING MODELS

## ABSTRACT

Reasoning models (e.g., DeepSeek-R1) use extra inference-time compute to write long chains of thought and solve harder problems. Yet they often loop—repeating the same text—especially at low temperatures or with greedy decoding. We take a step toward understanding why. We evaluate several open reasoning models and see looping is common at low temperatures. Within a family, higher capacity models loop less and for distilled models, the student loops far more even when the teacher rarely does. This points to imperfect learning or errors in learning as a key cause. We then demonstrate two ways errors in learning can cause loops, using a simple graph-traversal setup. First, when the correct next action is hard to learn but an easy cyclic action is available, the model puts relatively more probability on the easy action and gets stuck. Second, errors across time steps in a chain of thought can be correlated, which drives repetition. Finally, we discuss potential avenues for reducing looping and implications beyond looping.

## 1 INTRODUCTION

Reasoning models (Jaech et al., 2024; DeepSeek-AI et al., 2025; Abdin et al., 2025; Guha et al., 2025) use extra inference time compute, generating long chains of thought, to solve harder problems. This has opened a complementary scaling axis of inference-time compute, alongside training compute, resulting in striking gains on challenging tasks such as competitive math and coding. Yet these models often get stuck in loops: endlessly repeating the same text in their chain of thought, especially under greedy decoding and low temperatures (see Appendix D for an example). As a result, most model providers recommend running them at a sufficiently high temperature to avoid looping (e.g., see the Hugging Face pages for DeepSeek-R1 and QwQ-32B).

This raises several questions: Why do these models loop, and how does temperature help? Does temperature address the root cause or merely mask the symptoms? Is there a cost of high temperature? For instance, error accumulation across generation steps increases with temperature. Ideally, temperature would be a knob that controls the degree of exploration in a chain of thought rather than a stopgap for looping. More fundamentally, is randomness a necessary resource for good reasoning models? This is reminiscent of classical questions in algorithms on whether randomized algorithms are more powerful than deterministic ones (Motwani & Raghavan, 1996; Vadhan et al., 2012).

In this work, we take a step towards understanding these questions. Our contributions are as follows.

**Observations with open reasoning models (Section 2).** We evaluate several open reasoning models (e.g., DeepSeek–distilled Qwen, Openthinker-3, Phi-4 reasoning) for looping on problems from the American Invitational Mathematics Examination (AIME), a high-school math contest. We make several observations: (i) all models loop at low temperatures; (ii) within a family, smaller models loop more; (iii) for models trained via distillation, students loop far more than their teachers; and (iv) for most models, harder AIME problems elicit more looping. These observations point to imperfect learning—i.e., systematic *errors in learning* of the training distribution—as a key cause. If a student perfectly learned the teacher, then the amount of looping of the student cannot be significantly higher than the teacher. For instance (Figure 1), *how can it be that Openthinker3-1.5B loops in 26% of its responses with greedy decoding, while its base instruction-tuned model and teacher barely loop?*

**Modeling and understanding looping.** Next, we introduce a simple graph reasoning task with star graphs to isolate how errors in learning cause looping, building on Bachmann & Nagarajan

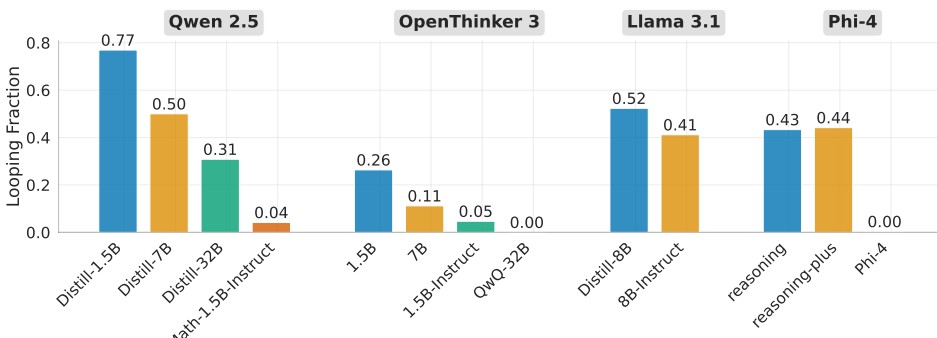

Figure 1: **Looping with greedy decoding (AIME 24/25 average).** Bars show the fraction of responses satisfying our $n$-gram looping criterion. All reasoning models loop at temperature 0 (except QwQ-32B which we show does still loop under a relaxed criterion). Within a family, larger models loop less (e.g., Qwen 1.5B > 7B > 32B). Distilled students can loop even when their teacher barely loops (OpenThinker3 vs. QwQ-32B). Instruction-tuned baselines generally loop far less than their reasoning counterparts. RL post-training has limited effect (Phi-4 Reasoning vs. Reasoning-Plus).

(2024). We train Transformers from scratch on random-walk traces that start at a designated start node and aim to reach a leaf goal—mimicking a chain of thought that sometimes makes progress and sometimes backtracks. We highlight two mechanisms:

*Hardness of learning (Section 3).* We show that when the correct progress-making action (e.g., the next step in a proof) is hard for a model to learn while an easy cyclic action is available (e.g., backtracking to a previous step), errors in learning lead to relatively more probability on the easy action. Under greedy decoding, that cyclic action is selected repeatedly, creating loops. We formalize this in Proposition 1, which shows that indistinguishability of the hard action diffuses its probability across many alternatives, whereas the easy action retains its mass. We demonstrate this in our graph reasoning task, showing low-temperature generation loops. Further, temperature reduces looping and improves accuracy here, but more as a stopgap: the model still assigns too little mass to the hard action, leading to longer-than-necessary chains.

*Temporally correlated errors (Section 4).* We show that even in the absence of hardness of learning, Transformers can have an inductive bias for repetition: when the training distribution places (nearly) equal mass on several progress-making actions, small estimation errors tilt the model toward a few options, and these errors correlate over time. When a similar decision point reappears later in the chain, the model tends to reselect the previously favored actions. Low-temperature decoding amplifies these small errors, leading to loops. We again demonstrate this in the graph reasoning task. In this case, however, we argue that higher temperature is a reasonable remedy—not a mere stopgap—because it smooths away small, correlated errors rather than masking a large probability gap.

**Broader implications and future directions (Section 5).** These results suggest that randomness plays the role of smoothing out learning errors, but whether it is a principled solution or a stopgap depends on the underlying mechanism. Finally, using insights from our analysis, we end with several directions that can lead to better reasoning models in general along with fixing looping.

## 1.1 RELATED WORK

While looping has been especially prevalent in *reasoning* models, it has been observed and studied since the early days of large language models. Holtzman et al. (2020) brought broad attention to this "neural text degeneration," showing that low-temperature sampling or beam search can yield generic and repetitive text. In response, several mitigations were explored. Unlikelihood training explicitly down-weights repeated or undesirable continuations (Welleck et al., 2020), and contrastive methods encourage more isotropic token representations, which reduces repetition (Su et al., 2022). A key data-centric insight was that model repetitions were correlated with repetitions in the training corpus (Li et al., 2023); as instruction-tuning data improved and models scaled, looping became less severe. Consistent with this view, later analyses found that the anisotropy observed in earlier models (e.g.,

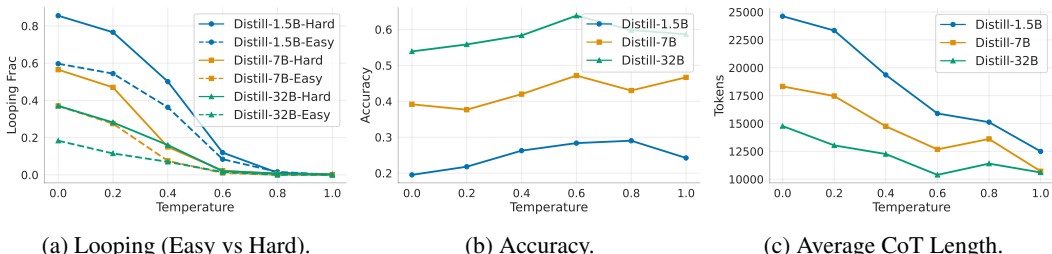

(a) Looping (Easy vs Hard).   (b) Accuracy.   (c) Average CoT Length.

Figure 2: **Response statistics for DeepSeek-R1–distilled Qwen (AIME24/25 average).** (a) Looping drops sharply with temperature; smaller models loop more, and hard problems induce more looping than easy ones. (b) Accuracy rises with temperature up to a mid-range peak ($\approx 0.6-0.8$); larger models are consistently more accurate. (c) Average chain-of-thought length decreases with temperature; even once looping subsides at higher temperatures, smaller models produce longer responses than larger models trained on the same data.

GPT-2) largely disappears in later families such as OPT (Su & Collier, 2023). This aligns with our evaluations as well: many instruction-tuned models we tested exhibit little looping.

With the rise of *reasoning* models, however, severe looping has re-emerged. The very nature of chain-of-thought data, which includes cyclic actions like backtracking and reflection (Li et al., 2025; DeepSeek-AI et al., 2025; Cuadron et al., 2025; Gandhi et al., 2025) provides fertile ground for models to amplify into degenerative loops. Moreover, scaling alone is not a satisfactory solution for reasoning models: a core promise of this paradigm is to leverage inference-time compute so that even small models can perform well via longer chains. Understanding and holistically mitigating looping in this setting is therefore important, and our work takes a step towards this.

## 2 OBSERVATIONS ON OPEN MODELS

We conduct a large-scale study of looping on openly available language models. This includes a range of model sizes and training paradigms like instruction tuning, distillation from a teacher reasoning model, and RL post-training. The reasoning models we tested are as follows. **Qwen:** DeepSeek-R1 Distilled Qwen 1.5B, 7B, 32B (DeepSeek-AI et al., 2025); **Openthinker3:** Open-Thinker3 1.5B, 7B (Guha et al., 2025) and QwQ-32B (Team, 2025), which is the teacher for OpenThinker-3 models; **Phi-4:** Phi-4-reasoning, Phi-4-reasoning-plus (Abdin et al., 2025); **Llama:** DeepSeek-R1 Distilled Llama 8B (DeepSeek-AI et al., 2025). The instruct variants (non-reasoning models) we test are: Qwen2.5-Math-1.5B-Instruct (Yang et al., 2024b), Qwen2.5-1.5B-Instruct (Yang et al., 2024a), Phi-4 (Abdin et al., 2024), Llama-3.1 8B Instruct (Grattafiori et al., 2024)

We consider a text response to contain looping if it contains any $n$-gram at least $k$ times. We choose $n = 30$ and $k = 20$ for all reasoning models. Note that large $n$ makes it a fairly strict requirement and we observe that qualitative trends are not sensitive to $k$ (e.g., see Figure 10 for corresponding plots with 3 times bigger $k$). Additionally, since instruct models produce shorter responses, we relax the looping definition to have $k = 10$ for them. All plots report averages over AIME 2024 and 2025. For each triple (problem, model, temperature) with temperature $\in \{0, 0.2, 0.4, 0.6, 0.8, 1.0\}$, we sample 10 independent responses and compute accuracy, looping percentage, and response length; we then average these quantities across problems.

In Figure 1 we present the looping percentages with greedy decoding for all evaluated models. We show accuracy, looping percentages and response lengths as a function of temperature for Qwen models in Figure 2 and for other model families in Appendix B. Our observations are:

**Models loop at low temperature.** We see this across all open reasoning models we tested on AIME. As temperature approaches 0, the amount of looping increases. One model that showed almost no looping for AIME was QwQ-32B. But even there, we do see looping with a relaxed definition. In fact, the model provider HuggingFace page explicitly says not to use low temperature to avoid endless repetitions (see Appendix D where we show other examples of looping for QwQ-32B). Further, we see that looping decreases with temperature (Figure 2a) and accuracy increases with temperature till a certain point (Figure 2b).

**Higher capacity models loop less.** This is evident from the Qwen and the Openthinker3 model families. Further, in the Openthinker3 case we observe that the distilled models have a considerable amount of looping, even though their teacher model (QwQ-32B) shows negligible looping. Also, while looping vanishes at higher temperatures, the higher capacity models still produce shorter responses (Figure 2c), despite having been trained on the same data. As we will argue in later sections, this phenomenon is closely related to that of looping.

**Harder problems elicit more looping.** We split the AIME problems into easy and hard by considering the first 5 problems of AIME I and first 5 of AIME II as easy, and the rest as hard (based on the hardness rating guide from the AoPS Wiki). We observe more looping on hard problems for almost every model family we tested (see Figure 2a for Qwen). The only exception being Phi-4 reasoning, which we discuss in Appendix B. This suggests that no matter how large the scale of a model, there always exists a problem hard enough that induces looping in the model. We leave a more thorough investigation of this conjecture to future work.

**Reasoning models loop even when their instruct counter-parts do not.** For the Qwen, Openthinker, and Phi-4 model families, we see that the base instruction tuned models do not loop, while their reasoning counterparts loop a lot. It is not always the case that the base instruct model does not loop, as we can see from Llama-3.1-8B-Instruct. But even there we observe an increase in the looping percentage of the respective reasoning model. The amount a model loops is likely a function of cyclic actions like restatement or backtracking in the training data. This view is supported by past work (Li et al., 2023). In subsequent sections, we discuss this more.

**Effect of RL training.** Phi-4 reasoning is one model family where we have access to both a distilled model (Phi-4-reasoning) and one that has been (lightly) tuned with RL (Phi-4-reasoning-plus). In this setting, we observe that the looping counts remain roughly the same across the two models.

**What causes looping?** Two patterns stand out in the above observations: looping decreases with model capacity, and distilled students can loop far more than their (larger) teachers. If a student perfectly learned its teacher distribution, we would not expect it to loop substantially more than the teacher. The gap thus points to imperfect learning—systematic errors in the learned distribution—as a key cause. Further, most models looping more on harder problems aligns with this view. In the next two sections, we discuss mechanisms through which errors in learning can drive looping.

## 3 HARDNESS OF LEARNING

In this section, we show how the hardness of learning can lead to looping. For this discussion, it is useful to keep in mind the distillation scenario where a student model is fine-tuned on reasoning chains generated by a teacher (we later discuss how the ideas extend beyond distillation). As a simple example, suppose at some step in the reasoning chain, the training data distribution has support over two actions: a progress-making action (e.g., the next logical proof step) and a cyclic action (e.g., backtracking and trying again). Assume the progress-making action is hard for the student to learn, while the cyclic action is easy. We show that even if the training distribution puts high probability on the progress action, the student can still place relatively more mass on the cyclic action. Thus, while greedy decoding under the training distribution would lead to progress, greedy decoding with the student tends to pick the cyclic action repeatedly and get stuck.

To make this idea concrete, we say an action is hard if the model cannot distinguish it from $n$ other actions. For instance, there may be a natural next step in the proof, but the model confuses it with $n$ other possibilities. Larger $n$ here implies a harder action. In this case, even if the training distribution assigns high probability on the hard action, the model, trained to maximize log-likelihood, diffuses that mass across the indistinguishable options. The easy action then ends up with relatively higher mass. We formalize this in the proposition below.

**Proposition 1.** *Consider the following task: there exist $n$ sets of contexts $C_1, \ldots, C_n$ which are equi-likely under the training distribution. And there are $n$ distinct "hard" actions $a_1, \ldots, a_n$, and an "easy" action $a_0$. For every context $c_i \in C_i$, the training distribution picks action $a_i$ with probability $(1-p)$ and $a_0$ with probability $p$. Now consider a learner that cannot distinguish between the $n$ hard actions or, in other words, it is constrained to ignore the context when deciding on the best action. Then the maximum log-likelihood solution for such a learner assigns probability $p$ to the easy action $a_0$ and probability $(1 - p)/n$ to the hard indistinguishable actions $a_i, \forall i \in \{1, \ldots, n\}$.*

We provide the proof in Appendix C. To appreciate the implications, note that as the action becomes harder (i.e., as $n$ increases), the probability assigned to it decreases as $(1-p)/n$, while the probability on the easy action remains $p$. Thus, for sufficiently large $n$, greedy decoding picks the easy action.

**Mapping to language models.** In a language model, an action can be viewed as a short span of tokens implementing a logical step (e.g., the next step in a proof). For a span $x_{t:t+k-1}$, the model's probability of that action given the prefix is $P_\theta(x_{t:t+k-1} \mid x_{<t}) = \prod_{i=t}^{t+k-1} P_\theta(x_i \mid x_{<i})$, so the log-probability of the span is the sum of the per-token log-probabilities. Because next-token training maximizes this sum over tokens, it also maximizes the log-probability of any such span. Consequently, if a progress-making step is hard (confusable with $n$ alternative spans), the model spreads its mass across those spans, reducing the learned probability on the intended span by a $1/n$ factor, while an easy cyclic span retains its mass. The proposition therefore applies directly to these token-span actions. Also, note that cyclic actions discussed above are widespread in reasoning models, in forms such as backtracking and reflection (Li et al., 2025) (see Appendix D for an example).

### 3.1 DEMONSTRATION WITH GRAPH REASONING

We demonstrate the looping mechanism discussed above in a graph reasoning task.

**The star graph.** We build on the hardness result of Bachmann & Nagarajan (2024), who train Transformers to find paths in a star graph. A star graph $G(n, \ell)$ is a directed graph with a *root* $r$ and $n$ simple "spokes," each a path of length $\ell - 1$ ending at a distinct leaf (Figure 3). Each training example is a sequence containing the edge list, a start node (the root $r$), a goal node $g$ (a leaf chosen uniformly at random), and a path from $r$ to $g$. Distinct instances are formed by randomly permuting node labels. Bachmann & Nagarajan (2024) show that Transformers trained with next-token prediction fail to learn the correct path.

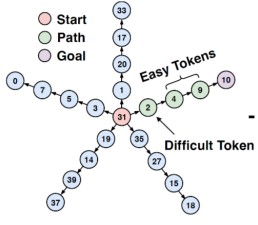

Figure 3: Illustration of star graph. Figure from Bachmann & Nagarajan (2024).

Here, aside from the root, all nodes have a single outgoing edge. Thus, for path-finding, learning the first edge on the path is harder than learning the remaining edges. Bachmann & Nagarajan (2024) show that models learn the later edges early in training; once those are learned, the first edge becomes the bottleneck. As a result, the learned solution places roughly $1/n$ probability on each outgoing edge from the root while predicting subsequent edges correctly, yielding chance accuracy. Recently, Hu et al. (2025) formalized this hardness and showed that, once the easy edges are learned, recovering the right solution is as hard as learning parity, which is conjectured to be hard for gradient-based optimizers (Abbe & Boix-Adsera, 2022; Shalev-Shwartz et al., 2017; Abbe & Sandon, 2023). Note that here the source of hardness is not model capacity (the models have enough capacity to represent the right solution), but the inability of optimization.

**Setup.** We make two modifications to the star-graph setting. *First*: instead of training on a single path from start to goal, we train on a random walk trace that begins at a start node. At any node, the walk moves forward to the next node on the path toward the goal with probability $1 - p$ (the progress-making action) and transitions back to the start node with probability $p$ (the backtracking/reset action). For simplicity, we apply this reset from every node, including the start node itself. When $p = 0$, this reduces to the original star-graph setting with a single path. We use $p = 0.3$.

*Second*: we introduce an explicit start node $s$ with a single outgoing edge to the root $r$. The root and leaf nodes remain as in the standard star graph, and the goal $g$ is still a leaf. This second modification is not crucial for our results, but it helps illustrate the mechanism better (see observations below). We abuse notation and use $G(n, \ell)$ to denote this modified star-graph where $n$ paths of length $\ell - 1$ emanate from the root, and there exists a separate start node with a single outgoing edge to the root. Finally, in some experiments, we also add a small exploration probability, described later.

$$\underbrace{11,16|5,42|2,29|...|29,22}_{\text{edge list}} / \underbrace{2,42}_{\text{start, goal}} = \underbrace{2,29,22,33,5,2,...,5,42}_{\text{random walk}}$$

An example training instance is shown above. Each instance is a sequence containing an edge list, a start node, a goal node, followed by a random walk from start to goal. Each node is a separate token, and $\{|, /, =\}$ are separator tokens. As in the original star-graph setting, distinct instances are formed by randomly permuting node labels.

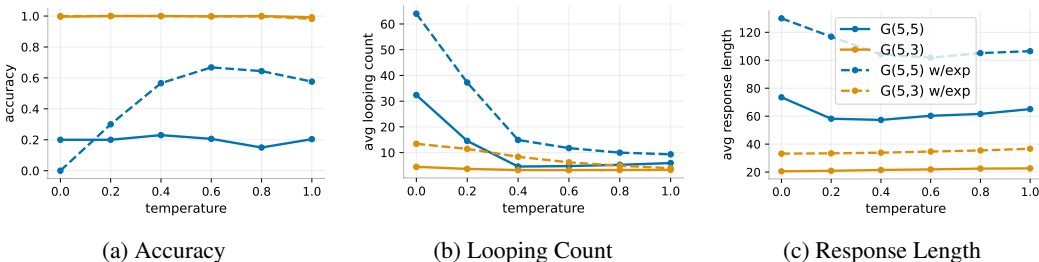

(a) Accuracy          (b) Looping Count          (c) Response Length

Figure 4: **Hard to learn actions induce low-temperature loops.** We train small Transformers from scratch on random-walk traces on star graphs $G(5,5)$ (hard) and $G(5,3)$ (easier), with progress-making probability $0.7$ and cyclic/backtracking reset $p = 0.3$. Dashed curves use an exploration variant at the root ($0.5$ correct edge, $0.2$ other paths, $0.3$ reset). The looping count is the average number of root→start transitions per trace. On the hard graph, looping is large at low temperature and falls as temperature increases; the average response length shows the same trend ((b),(c)). On the easy graph, both quantities are small and nearly flat. Accuracy (a): for $G(5,5)$ without exploration, accuracy stays near chance across temperatures; with exploration it rises with temperature but is near 0 at $T{=}0$. $G(5,3)$ remains near-perfect at all temperatures. Overall, a hard progress-making decision at the root, paired with an easy cyclic action, drives low-temperature loops.

If one views the source-to-goal path in the original star-graph setting as a simple model for the chain of thought of earlier LLMs that moves towards the goal in a step-by-step manner, then the random-walk variant can be a model for reasoning LLMs that explore multiple strategies, backtrack, and restart. Having said that, our aim with this setting is not to capture the complete complexity of an actual reasoning LLM, but to isolate the phenomenon of interest in the simplest setting possible.

**Training Details.** We train a decoder-only Transformer from scratch, with 12 layers, 8 attention heads, and 768 embedding dimension ($\approx 85M$ parameters). We use Adam for 100k steps with a learning rate of $10^{-4}$ (cosine decay) and batch size 64. We use 2M training sequences and train with cross-entropy loss for next-token prediction, applying the loss only to the random-walk portion of the sequence. At test time, the model receives new randomly generated instances and, given the edge list, start, and goal, is expected to generate a walk from the start to the goal. We mark a generated walk as accurate if it takes only valid transitions and eventually reaches the goal and stops.

## 3.2 OBSERVATIONS.

**Low temperature looping.** The random walk only visits nodes along the path from the start to the goal. At each visited node, the training distribution places probability $1 - p = 0.7$ on the *progress-making* action and $p = 0.3$ on the *cyclic/backtracking* action (reset to the start). A perfect learner would, under greedy decoding, always take the progress-making action and reach the goal without looping. However, hardness at the root breaks this behavior and induces looping.

The reset action is easy to learn: it only requires reading the start node from context and jumping to it. The progress-making action is also easy at all nodes except the root, where the model must choose among $n$ outgoing edges. If it cannot distinguish these $n$ options, it spreads the $0.7$ mass roughly uniformly, assigning $\approx 0.7/n$ to each, while still assigning $0.3$ to the reset. Under greedy decoding (or low temperature), this means the model chooses reset whenever $0.3 > 0.7/n$. The result is a two-node loop between the root and the start. Note that the model does not get trapped in the start node's self-transition: both "move forward" and "stay/reset" at the start are easy to learn, so it typically assigns these actions probability close to 0.7 and 0.3 respectively, as intended. Loops arise when a hard action (root fan-out) coexists with an easy cyclic action (reset). We added the explicit start node in our setup to demonstrate this.

We observe this behavior on $G(5,5)$. Figure 4b shows the *average looping count*, defined as the average number of root to start transitions per test instance. The count is high at low temperature and decreases as temperature increases. Figure 4c shows that the average response length is also larger under greedy decoding and decreases with temperature. At temperature 0, the looping count is roughly half the response length, since traces often alternate start $\rightarrow$ root $\rightarrow$ start $\rightarrow$ root $\cdots$, and we count only the root $\rightarrow$ start transitions. Finally, although the training process is Markovian at

each step, the learned model is not exactly Markovian; after bouncing between the start and the root, it eventually commits to a path but, lacking the correct outgoing edge from the root, wanders to an arbitrary leaf. As a result, accuracy is near chance (Figure 4a).

**Accuracy increases with temperature.** While both looping count and response length drop with temperature, accuracy stays near chance across temperatures. This contrasts with many reasoning models, where accuracy often improves as temperature increases. The difference stems from the training distribution: the walk never explores off-path routes, so the model learns to revisit the *same* path after each reset. Since it cannot reliably pick the correct path at the root, accuracy remains at chance even when loops shorten.

To test exploration, we modify the walk at the root: with probability $0.3$ the walk resets to the start as before, with probability $0.5$ it takes the correct outgoing edge (progress-making action), and with probability $0.2$ it takes one of the other paths uniformly at random (exploration). At all other non-leaf nodes (out-degree 1), the walk moves forward with probability $0.7$ and resets with probability $0.3$; it stops at the goal and, upon reaching a non-goal leaf, resets with probability $1$. Training on such traces increases accuracy with temperature on $G(5, 5)$ (w/exp in Figure 4). Interestingly, accuracy at temperature $0$ drops near $0$: the model tends to bounce between start and root and terminate eventually (e.g., by emitting EOS) rather than committing to a path. Looping count and response length still decrease with temperature, but both are higher than in the non-exploration setting, likely because exploration yields longer training traces and the model mirrors the increase at test time.

**Less looping on easier problems.** We also evaluate $G(5, 3)$ (with and without exploration). Because the paths emanating from the root are shorter, the progress-making action at the root is less hard. Indeed, the learned probabilities show higher mass on the correct outgoing edge from the root than on the others. Early on, the model may still slightly prefer the reset at the root, so brief looping occurs, but it quickly takes the correct edge and reaches the goal. As a result, accuracy is near perfect across temperatures, and the average response length is stable. This mirrors our earlier finding from Section 2: models loop less on easier problems.

## 3.3 OTHER IMPLICATIONS

**Temperature as a mitigation.** Increasing temperature reduces looping and improves accuracy, but it does not remove the underlying hardness: the model still assigns too little probability to the correct progress-making action. A simple diagnostic is response length at high temperature. On $G(5, 5)$ with exploration, the learned model at temperature $1$ produces an average length of $106.5$, whereas a perfect learner would have an average length of $25.8$, which is more than 4x shorter. Thus temperature helps by exploring, not by correcting the probability shortfall, and the generations remain much longer than necessary. This also explains why, even at higher temperatures, smaller reasoning models tend to produce longer chains than larger models or their teachers. More holistic fixes would require training-time interventions; we return to these in Section 5.

**Sources of hardness.** In the illustration above, hardness comes from the inability of the optimization to find the right solution. More generally, hardness can arise due to other factors too such as limited model capacity or limited training compute (under-training). The hardness due to model capacity is a plausible explanation for why smaller models loop more within a family trained on the same data.

**Instruct vs Reasoning Models.** Two ingredients are needed in the mechanism above: (i) hard-to-learn actions, along with (ii) easy-to-learn cyclic actions present in the training distribution. The presence of hard actions amplifies the frequency of easy cyclic actions in model generations. However, if cyclic actions are rare in the training traces, extensive looping is less likely. This is a plausible explanation for why many instruction-tuned models loop less than reasoning models as reasoning traces include more cyclic actions such as re-statement and backtracking.

**RL vs distillation.** Note that the mechanism discussed in this section can affect models trained via RL too, and not just distillation. A guiding principle here is that whenever there is a capacity difference between the teacher model and the student model, hardness of learning for the student can amplify looping behavior. One can approximately view the RL training process as sampling multiple trajectories from the model and training on the correct ones. While there is no explicit teacher model for a RL trained model, this can be thought of as training the model on best-of-$k$ version of itself where $k$ is the number of trajectories sampled. In that sense, there is still a gap between the data generator and the learner, which can possibly cause this mechanism.

# 4 CORRELATED ERRORS ACROSS TIME-STEPS

In this section, we describe another mechanism by which errors in learning cause looping. It is easiest to see in the graph reasoning setting, so we directly jump in.

**Setup.** We use the same star graph as in Section 3 (including the start node), but change the random walk. The walk begins at the start node. At each non-leaf node, it chooses one outgoing edge uniformly at random. Thus all non-root internal nodes (out-degree 1) always take their unique edge, while the root chooses uniformly among its $n$ children. If the walk reaches the goal (a leaf), it stops; if it reaches a non-goal leaf, it transitions back to the start. We train Transformers on traces drawn from this process; training and test details match the previous experiment.

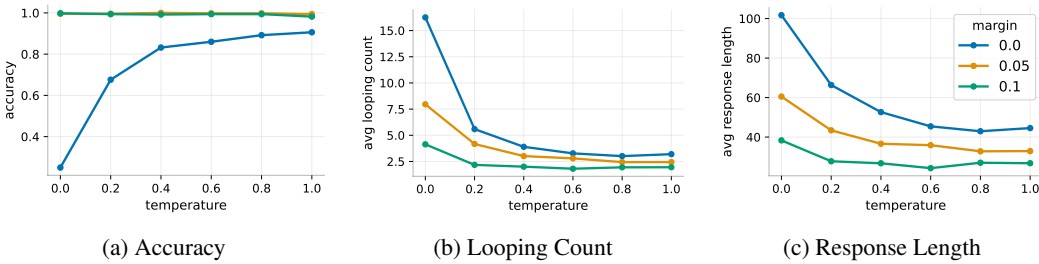

|                |                  |                     |
| :------------: | :--------------: | :-----------------: |
| (a) Accuracy   | (b) Looping Count | (c) Response Length |

Figure 5: **Temporally correlated errors induce low-temperature loops.** We train on $G(5,5)$ star-graph random-walk traces that choose a child uniformly at the root, take the unique outgoing edge at other internal nodes, and reset to the start at non-goal leaves. The learned model makes *temporally correlated* errors at the root, reselecting a small subset of edges at low temperatures and thereby looping; this reduces—but does not vanish—when training incentivizes visiting new children via a margin ($m \in \{0, 0.05, 0.1\}$). (a) *Accuracy:* near chance for $m{=}0$ at low temperature and increasing with temperature; margin variants achieve near-perfect accuracy across temperatures. (b) *Looping count:* for each trace, count visits to each root child and take the maximum; we plot the average over test instances. Lower temperatures have higher looping counts; margins reduce them. (c) *Response length:* longer at lower temperatures, and shortened by margins.

## 4.1 OBSERVATIONS AND IMPLICATIONS.

**Low temperature loops.** It helps to first consider a perfect learner. On $G(n,\ell)$, such a model would learn: probability 1 to the unique outgoing edge at non-root internal nodes; probability 1 to reset at non-goal leaves; and probability $1/n$ to each child at the root. In practice, the trained model deviates slightly at the root. Instead of learning an exact $1/n$ split, it makes small errors (e.g., $0.2 \pm 0.05$ for $n{=}5$). More importantly, these errors are *correlated across time*: the root children that are slightly preferred early in the trace tend to remain preferred on later visits to the root. Under greedy decoding or low temperature, the model therefore keeps revisiting the same one or two paths, producing loops.

We quantify this with a *looping count*: for each generated trace, we record how many times each root child is visited and report the maximum over children; we then average this value over test instances. In Figure 5 (margin $= 0$), we show the accuracy, looping count and average response length versus temperature for a model trained on $G(5,5)$. We observe that the looping count is high at low temperature and decreases as temperature increases. The average response length shows the same trend. Accuracy is near chance at temperature $0$ and improves with temperature.

**A variant with margins.** The training walk above samples root children uniformly at every visit. What if the training distribution itself discourages revisiting already-seen children? We study a *margin* variant: the first time the root is visited, a child is sampled uniformly; on later visits, each previously visited child has its sampling probability reduced by a fixed margin $m$, and the removed mass is redistributed uniformly over the as-yet-unvisited children. Note that with $m{=}0$ we recover the random walk considered above. Also, for $m{>}0$ the process is no longer Markovian.

Training with $m{=}0.05$ and $m{=}0.1$ reduces low-temperature looping counts and response lengths compared to $m{=}0$, and accuracy becomes near-perfect across temperatures. However, looping does not vanish: there remains a noticeable gap between temperature $0$ and higher temperatures. There are two ways to interpret this: On one hand, it shows the robustness of the looping mechanism—

the student still over-prefers already-visited children even when the training distribution nudges it away. On the other hand, this suggests a mitigation: biasing traces toward *new* actions reduces looping. While this is a training-time intervention in these experiments, existing inference interventions explicitly discouraging repetition are in a similar vein (Keskar et al., 2019).

**Interpretation.** Two ingredients drive this phenomenon: (i) the training distribution at certain decision points (e.g., the root) remains roughly the same across multiple points in a trace, and (ii) the learned model's errors at those points are temporally correlated. As a result, the training traces look reasonable, exploring diverse actions. However, the learned model at low temperature tends to reselect the same few actions because these errors persist, creating loops. Mapping to language models in a distillation setting: imagine the teacher spreads probability mass across several plausible strategies at multiple points in the chain; when sampled, it explores broadly, but the student inherits temporally correlated preferences and, at low temperature, repeats a few of them in a loop.

**Comparing the two mechanisms.** Both looping mechanisms we discuss stem from errors in learning, but they differ in nature. The first arises from *hardness of learning*: probability mass on a hard progress-making action is diffused across many indistinguishable alternatives. Importantly, this gap can be large relative to the training distribution and does not rely on Transformer-specific inductive biases—it appears for any maximum-likelihood learner when the correct action is indistinguishable from many others. The second relies on an *inductive bias toward temporally correlated errors*: the learned probabilities at repeated decision points are slightly but consistently skewed toward a few actions. Here the deviations are small, yet sufficient for greedy/low-temperature decoding to loop.

**Temperature as a mitigation.** In this mechanism the probability errors are small, so increasing temperature effectively smooths them out: loops shrink and accuracy rises. The contrast with Section 3 is instructive. There, the hard progress-making action receives far too little mass, so temperature helps only by occasionally sampling it—chains remain long ($\approx 4\times$ response length compared to the perfect learner). Here, the gap is modest: for example, on $G(5,5)$ with margin $= 0$, the learned model at temperature 1 has average response length $44.5$ compared to $28.6$ for the perfect learner ($\approx 1.5\times$ blowup). In short, temperature is a good fix when the learned probabilities are close but slightly skewed; when the errors are large, training-time interventions are a more holistic fix.

## 5 DISCUSSION

We began with a simple question: why do reasoning models loop, and is temperature a real fix or a stopgap? Our evaluation of open reasoning models points to *errors in learning* as a central cause. In controlled settings we then showed two mechanisms through which errors drive loops, and explained behavior seen in the reasoning model evaluations. We found that temperature serves the purpose of smoothing out learning errors. It is a reasonable fix when the errors are small (e.g., temporally correlated biases), but it is merely a stopgap when errors are large due to hardness of learning.

Note that hardness is especially salient when distilling large reasoning models into small ones—the dominant way of training small reasoning models today. Thus fixing these errors holistically is important for training high-quality small reasoning models, beyond reducing looping. A concrete direction here is targeted data augmentation: identify points in teacher traces which the student finds hard to learn (e.g. high loss) and augment them with brief hints. Other levers include better curricula and architectures to mitigate hardness (e.g., for stargraph style hardness, recent work shows newer architectures can help (Hu et al., 2025; Ahn et al., 2025)).

Among inference-time interventions, adaptively choosing the temperature is another promising direction. One clear downside of temperature is error-accumulation. If one can find points during generation where using temperature helps smooth out errors, using temperature selectively at those points can improve model generations.

**Limitations.** Our controlled demonstrations are intentionally simple to isolate mechanisms behind looping. Real tasks are messier and other forces beyond what we discuss likely matter. Still, the patterns we highlight align with behavior in open models. Our work is just a step towards a better understanding of looping in reasoning models, and not the last word. We hope it spurs future research on better diagnostics, stronger distillation methods, and more selective generation methods—ultimately yielding more efficient and accurate reasoning models.

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

# A    LLM USAGE.

We used LLMs for basic editing and rephrasing.

# B    ADDITIONAL EXPERIMENTAL DETAILS ON SECTION 2

**Evaluation details.** We used the Eureka ML Insights Framework (Balachandran et al., 2024; 2025) to conduct the evaluations on LLMs. We ran all reasoning models with a 30K `max_tokens` budget and all base non-reasoning models with 3K `max_tokens`. For each problem and each temperature, we computed the statistics by averaging over 10 different sampled responses from the model. The numerical answer for each problem was extracted from the part of the response after the end of thinking (typically denoted with the `</think>` token.) In terms of compute, we used a node with eight B200 GPUs and each model evaluation took about a day to complete, on average.

**Phi-4 reasoning models on hard and easy problems.** The Phi-4 reasoning family of models was the only exception to the observation that hard problems induce more looping than easy ones. As we can see in Figure 7d, both phi-4-reasoning and phi-4-reasoning-plus consistently exhibit more looping in easier problems than in hard ones. By manually inspecting its responses, we realized that it very often demonstrates a peculiar form of looping; it finds the correct answer during the CoT, then proceeds to present it, and then it gets stuck indefinitely repeating things like "We'll produce answer in plain text." This is one of the primary ways in which it loops, which means that for easy problems, it will reach the solution more frequently and, thus, get stuck in this situation more often than in harder problems. Note also that another key way in which phi-4-reasoning models differ from the other models we tested is that it has been fine-tuned on OpenAI o4-mini data (Abdin et al., 2025), as opposed to DeepSeek-R1 or Qwen. Nevertheless, this is only a preliminary attempt at explaining this discrepancy and a more thorough study of the exact underlying factors would make a great direction for future research.

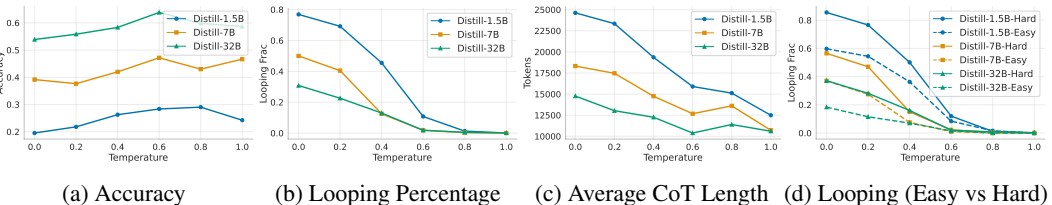

    (a) Accuracy    (b) Looping Percentage    (c) Average CoT Length    (d) Looping (Easy vs Hard)

Figure 6: Qwen metrics as a function of temperature.

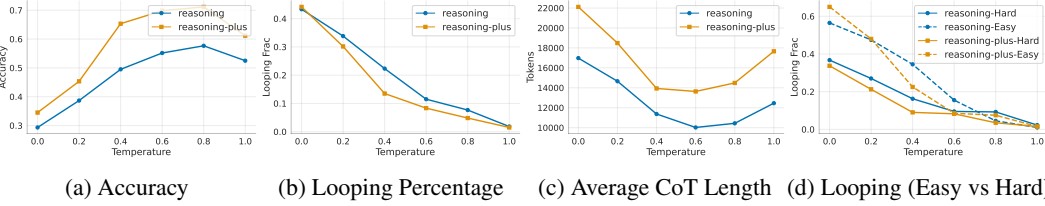

    (a) Accuracy    (b) Looping Percentage    (c) Average CoT Length    (d) Looping (Easy vs Hard)

Figure 7: Phi-4 metrics as a function of temperature

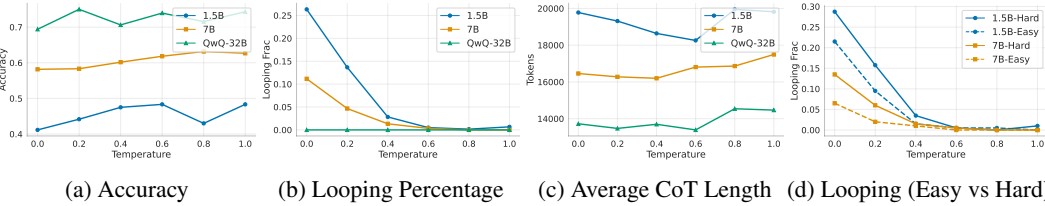

    (a) Accuracy    (b) Looping Percentage    (c) Average CoT Length    (d) Looping (Easy vs Hard)

Figure 8: Openthinker metrics as a function of temperature.

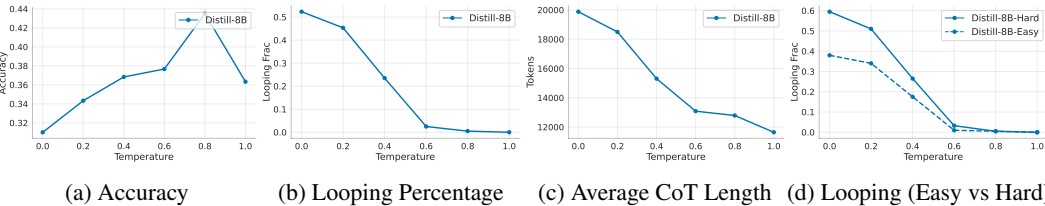

(a) Accuracy  (b) Looping Percentage  (c) Average CoT Length  (d) Looping (Easy vs Hard)

Figure 9: Llama metrics as a function of temperature.

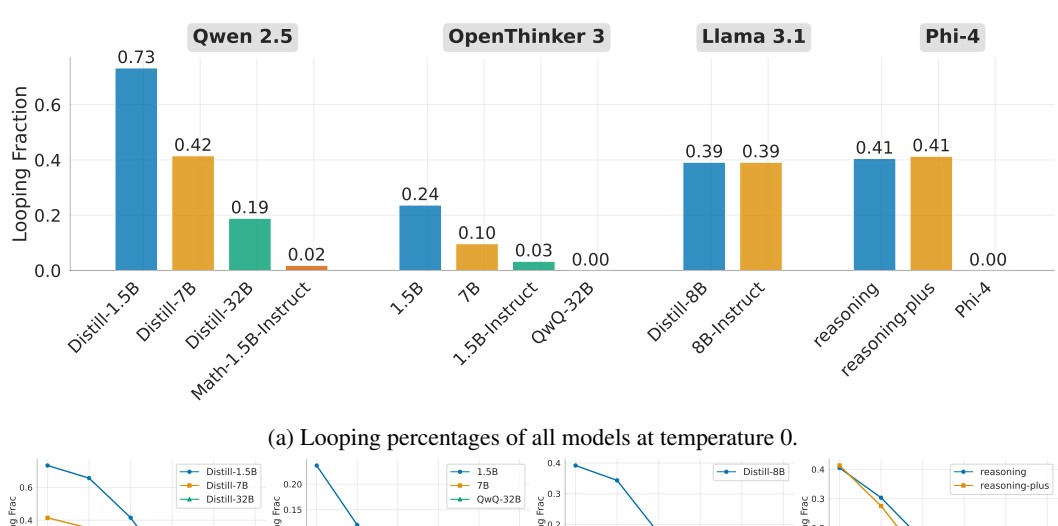

(a) Looping percentages of all models at temperature 0.

(b) Qwen Looping  (c) Openthinker Looping  (d) Llama Looping  (e) Phi-4 Looping

Figure 10: Looping percentages if we increase the $n$-gram threshold to $k = 60$.

## C  PROOF OF PROPOSITION 1

We restate the proposition below and provide its proof.

**Proposition 1.** *Consider the following task: there exist $n$ sets of contexts $C_1, \ldots, C_n$ which are equi-likely under the training distribution. And there are $n$ distinct "hard" actions $a_1, \ldots, a_n$, and an "easy" action $a_0$. For every context $c_i \in C_i$, the training distribution picks action $a_i$ with probability $(1-p)$ and $a_0$ with probability $p$. Now consider a learner that cannot distinguish between the $n$ hard actions or, in other words, it is constrained to ignore the context when deciding on the best action. Then the maximum log-likelihood solution for such a learner assigns probability $p$ to the easy action $a_0$ and probability $(1 - p)/n$ to the hard indistinguishable actions $a_i, \forall i \in \{1, \ldots, n\}$.*

*Proof.* Let $q_1, \ldots, q_n$ be the probabilities that the model assigns to the $n$ actions, and $q_0$ be the probability of the easy action. Then, the total Cross-Entropy loss is

$$\ell = -p \log(q_0) - (1-p)\frac{1}{n}\sum_{i=1}^{n} \log(q_i).$$

By taking derivatives and setting them to 0 we find that the stationary point is achieved at $q_0 = p$ and $q_i = (1 - p)/n$ for every $i = 1, \ldots, n$. This in turn is the minimizer due to the convexity of Cross-Entropy. ☐

# D  LOOPING EXAMPLES

We give examples of cases where the model loops during its response.

---

**DeepSeek-R1-Distill-Qwen-1.5B**

**Prompt (AIME 2025 II, Problem 12):**

Problem: Let $A_1 A_2 A_3 \\ldots A_{11}$ be an $11$-sided non-convex simple polygon with the following properties:\n\n\\begin{itemize}\n\\item For every integer $2 \\le i \\le 10$, the area of $\\triangle A_i A_{1} A_{i+1}$ is equal to $1$.\n\\item For every integer $2 \\le i \\le 10$, $\\cos(\\angle A_i A_{1} A_{i+1}) = \\\frac{12}{13}$.\n\\item The perimeter of the $11$-gon $A_1 A_2 A_3 \\dots A_{11}$ is equal to $20$.\n\\end{itemize}\n\nThen $A_1 A_2 + A_1 A_{11} = \\frac{m \\sqrt{n} - p}{q}$ where $m, n, p$, and $q$ are positive integers, $n$ is not divisible by the square of any prime, and no prime divides all of $m, p$, and $q$. Find $m + n + p + q$.\nMark your solution with \\boxed\nAnswer:

**Model Response:**

<think>Okay, so I have this problem about an 11-sided non-convex simple polygon, which is a bit intimidating because it's not convex, but it's also a polygon with specific properties. Let me try to break it down step by step.\n\nFirst, the polygon is labeled \\( A_1 A_2 A_3 \\ldots A_{11} \\). It's non-convex, which means it has at least one interior angle greater than 180 degrees, so it's not just a regular polygon. But it's a simple polygon, meaning it doesn't intersect itself.
...
So, using the Law of Cosines on triangle \\( \\triangle A_i A_1 A_{i+1} \\):\n\n\\( A_i A_{i+1}^2 = a^2 + a^2 - 2 a^2 \\cos \\theta_i \\)\n\nSo, \\( A_i A_{i+1}^2 = 2 a^2 (1 - \\cos \\theta_i) \\)\n\nGiven \\( \\cos \\theta_i = \\frac{12}{13} \\), so \\( 1 - \\cos \\theta_i = \\frac{1}{13} \\). Therefore,\n\n\\( A_i A_{i+1}^2 = 2 a^2 \\times \\frac{1}{13} = \\frac{2 a^2}{13} \\)\n\nSo, \\( A_i A_{i+1} = \\sqrt{\\frac{2 a^2}{13}} = \\frac{a \\sqrt{2}}{\\sqrt{13}} = \\frac{a \\sqrt{26}}{13} \\)\n\nSo, each side \\( A_i A_{i+1} \\) is \\( \\frac{a \\sqrt{26}}{13} \\).
...
Wait, so perhaps the problem is that I'm misapplying the Law of Cosines. Maybe the two equal sides are not \\( A_1 A_i \\) and \\( A_1 A_{i+1} \\), but rather \\( A_1 A_i \\) and another side.\n\nWait, but in that case, the two equal sides would be \\( A_1 A_i \\) and \\( A_1 A_{i+1} \\), which would make all sides from \\( A_1 \\) to \\( A_{11} \\) equal, which again leads to the contradiction.\n\n
Wait, so perhaps the problem is that I'm misapplying the Law of Cosines. Maybe the two equal sides are not \\( A_1 A_i \\) and \\( A_1 A_{i+1} \\), but rather \\( A_1 A_i \\) and another side.\n\nWait, but in that case, the two equal sides would be \\( A_1 A_i \\) and \\( A_1 A_{i+1} \\), which would make all sides from \\( A_1 \\) to \\( A_{11} \\) equal, which again leads to the contradiction.\n\n
... [repeated x110 times]

---

We also investigate looping behaviors in the COTs of the QwQ-32B model. We sourced prompts and chains from the OpenThoughts dataset Guha et al. (2025). To systematically find looping COTs, we use a rolling hash detection of repeating $n$-grams. In particular, we flag COTs that have $n$-grams of size $n = 30$, repeating more than 20 times, and then manually inspect the results. We present here a case of looping QwQ-32B COT.

918
919
920
921
922
923
924
925
926
927
928
929
930
931
932
933
934
935
936
937
938
939
940
941
942
943
944
945
946
947
948
949
950
951
952
953
954
955
956
957
958
959
960
961
962
963
964
965
966
967
968
969
970
971

---

**QwQ-32B**

**Prompt:**

There is a rooted tree of N vertices rooted at vertex 1. Each vertex v has a value A_{v} associated with it. You choose a vertex v (possibly the root) from the tree and remove all vertices on the path from the root to the vertex v, also including v. This will result in a forest of zero or more connected components. The beauty of a connected component is the \mathrm{GCD} of the values of all vertices in the component. Find the maximum value of the sum of beauties of the obtained connected components for any choice of v. Here, \mathrm{GCD} stands for [Greatest Common Divisor].
...

**Model Response:**

```
<think>
Okay, I need to solve this problem where I have a rooted tree,
and when I remove a path from the root to some vertex v, I
split the tree into several connected components. The beauty
of each component is the GCD of all its vertices' values,
...
Hmm, so how do we compute this efficiently? First, I need to
compute the path from root to each node v.
...
Wait, the approach might be: For a given v's path
(r → p → q → ... → v): Each node except the root has a parent
in the path.
...
Wait, no, but if we precompute for each u the
...
Wait, the next for v is nothing, since v is the end. Hence the
total sum for a path would be:
sum_total = 0
for each node u along path (including v) except v:
  contribution = sum_children_gcd[u] – G(u → next_node_in_path)
  sum_total += contribution
Wait but wait, also for the node v, its contribution would be
subtracting the last edge (from v's parent to v), but the edge
to the parent is already part of the
...
But in Python, loops can be handled. Now, considering that
even in N=3e5, a few loops are O(N) steps. Therefore this code
should work. Thus, the approach is correct.
</think>
```python
import sys import sys import sys import sys import sys
import sys import sys import sys import sys import sys
import sys import sys import sys import sys import sys ...
```

---

