# OpenReview forum: "Let's Let's Let's Let's... Understand Looping in Reasoning Models"
_ICLR.cc/2026/Conference — Submitted to ICLR 2026_

### Official Review · Reviewer_Lw8Q · 2025-10-30

**Soundness:** 3
**Presentation:** 4
**Contribution:** 3
**Rating:** 6
**Confidence:** 3

**Summary:**

This paper contributes a systematic investigation of repetition phenomena in large language models (LLMs) and large reasoning models (LRMs): (1) It first contributes an empirical evaluation of how different factors of the model impact the likelihood of repetition; (2) It then introduces a theoretical analysis by studying the hardness of learning by traversing step graphs.

**Strengths:**

- The paper is well written.  I believe the related work section is especially well written. It first introduces the benefits of instruction-fine tuning. However, LRMs (which can also be seen as instruction-fine-tuned) are worse at controlling repetition. The writing has strongly motivated the paper. All figures are also clearly presented throughout the paper.
- The paper has an extensive workload. It has evaluated a series of open-source models over a series of factors. These sound experiments will add to the valuable knowledge of this field.
- I am not familiar with learning theory, but I believe it is creative to use the star graph traverse problem to study the hardness of learning. The authors also train transformer models from scratch to study the star graph.

**Weaknesses:**

The major weakness is that the paper lacks a formal linking between the star graph and the real-world reasoning problem (e.g. math reasoning) that the paper studied.
- Can the authors formalize the real-world reasoning problems as the star graph?
- If not, can the authors show a proof sketch for how real-world reasoning problems can be *reduced* to the star graph (in the theoretical CS way)?
- If neither is possible, can the authors further abstract the two problems and show their links?

I would give 8 to the empirical aspect of this paper and only 5 to the theoretical part. So now I am rounding to 6 and would like to hear from the authors.

**Questions:**

The following points are NOT weaknesses. I only take this opportunity to have research discussions with the authors.

Q1: Are there any other methods studying the repetition problem, which corroborates your findings?  How about other RL methods (e.g. MTCS-style reasoning) which might mitigate repetition? Aside from that, have you read the work from Ryan Cotterell's group that sees LMs as a Turing machine / finite state machine (if my memory is correct)? Because the paper studies state transition, I believe some discussions on the theoretical CS would add more value.

Q2: What is the difference between repetition in standard text generation, and reasoning generation? What characteristics of reasoning make the repetition more (or fewer)?

Q3: It is not compulsory, but why didn't the authors evaluate close-source models?

---

> ### Author Response · Authors · 2025-11-20
>
> Thank you for the detailed and thoughtful review, and for the many interesting follow-up questions. We feel encouraged that you found our experiments extensive and the star-graph setup creative. Your main concern was around the mapping between our controlled setup and real-world reasoning problems. In the response below, we clarify this mapping, add concrete example traces from real models to ground the mechanisms, and discuss your follow-up questions. We hope these additions help address your concerns, and we look forward to hearing from you.
>
> # Clarifying the mapping
>
> In the graph reasoning task, the model starts at the source node and does a random walk with the aim of reaching the goal node. At any node in the graph, it takes an action which is simply the choice of next node to pick. A cyclic action is an action that takes the walk to an already visited node, and a progress-making action is an action that moves the walk closer to the goal node.
>
> The analogous notions for reasoning language models can be defined at the level of short spans of tokens (``chunks’’). A chunk is a span of tokens that implements a single logical step in the chain-of-thought (for example, a line or small paragraph). Conditioned on the prefix generated so far, a language model induces a distribution over possible next chunks; these are the actions in our abstraction. So a reasoning language model can be thought of as doing a random walk, where at each step it conditions on the chunks generated so far, and samples a new chunk. A cyclic action corresponds to a chunk that repeats something already done before (e.g., restating a previously stated fact), while a progress-making action corresponds to a chunk that moves closer to the final goal (e.g., taking the next step in the proof or narrowing down the candidate set).
>
> However, we note that with the graph reasoning task, our goal is not to capture the full complexity of reasoning language models, but just to have enough complexity so as to be able to make progress on understanding the phenomenon of looping and the mechanisms behind it.  Further, in the next point, we discuss an additional study we did, where we go over some example traces, and see that they indeed exhibit elements of the proposed mechanisms.
>
> # Example traces
>
> To further understand the mapping between our mechanisms and actual reasoning models, we examined some looping reasoning traces from the OpenThinker-3 1.5B model. We discuss an example trace below.
>
> The model is solving the following AIME 2025 problem:
>
> > The twelve letters $A$, $B$, $C$, $D$, $E$, $F$, $G$, $H$, $I$, $J$, $K$, and $L$ are randomly grouped into six pairs of letters. The two letters in each pair are placed next to each other in alphabetical order to form six two-letter words, and then those six words are listed alphabetically. For example, a possible result is $AB,CJ,DG,EK,FL,HI$. The probability that the last word listed contains $G$ is $\\frac{m}{n}$, where $m$ and $n$ are relatively prime positive integers. Find $m + n$.
>
> First, the trace restates the problem and correctly counts the total number of pairings:
>
> >First, let me make sure I understand the problem correctly. We have the letters A, B, C, D, E, F, G, H, I, J, K, and L. These are to be randomly paired up into six pairs. Each pair is then arranged in alphabetical order to form a two-letter word. Then, all these six words are listed alphabetically. The question is asking for the probability that the last word in this list contains the letter G.
>
> >…
>
> > Let me start with the total number of ways to pair the letters. Since the letters are being paired into unordered pairs, the number of ways to partition 12 letters into 6 pairs is given by (12)! divided by (2^6 * 6!)....

---

> ### Author Response · Authors · 2025-11-20
>
> Next, it tries to reason about how the two-letter words are formed and loops (semantically) for a bit about alphabetical ordering of pairs:
> > Each pair is formed by taking two letters and putting them in alphabetical order. So, for any pair, say, (X, Y), if X comes before Y in the alphabet, then the word is XY; otherwise, it's YX. Wait, but actually, the problem says "the two letters in each pair are placed next to each other in alphabetical order to form six two-letter words". So, regardless of the order in the pair, they are arranged in alphabetical order. So, for any pair, the two letters are ordered such that the first is before the second in the alphabet. Therefore, each pair is effectively an ordered pair in the alphabetical order. So, for example, if the pair is (C, D), the word is CD. If the pair is (D, C), the word is DC, but since they are placed in alphabetical order, it's DC? Wait, no, actually, if the pair is (D, C), then since D comes after C, they would be placed as CD? Wait, no, hold on. Wait, the problem says "the two letters in each pair are placed next to each other in alphabetical order". So, regardless of the original order of the pair, they are arranged in alphabetical order. So, for any pair, the two letters are sorted, so the first letter is the earlier one, the second is the later one. So, for example, if the pair is (C, D), the word is CD. If the pair is (D, C), the word is DC. Wait, but actually, the pair is just two letters, so the pair is unordered, but when forming the word, they are arranged in alphabetical order…
>
> Later, the model comes out of this semantic loop and outlines a high-level counting strategy. After that, it again falls into a loop, repeatedly re-deriving the same fact about sorted pairs, and never gets out of it:
>
> > Therefore, to compute this probability, we can think of all possible pairings, and count how many pairings result in the maximum word in the set containing G, then divide by the total number of pairings.
>
> > Alternatively, maybe it's better to think in terms of the possible pairs that include G and how they can be arranged.
>
> > First, let me note that G is the 7th letter. So, the letters before G are A, B, C, D, E, F, and after G are H, I, J, K, L. So, G is in the middle. So, the pairs that include G can be either (G, something before G) or (something before G, G). But when sorted, the pair with G will be either (G, something before G) or (something before G, G) but sorted, so it will be (something before G, G). Wait, no. Wait, if the pair is (G, H), then the word is GH. If the pair is (H, G), the word is HG, but since they are sorted, it's GH. Wait, so actually, the word is always the two letters in alphabetical order, so the word is the pair sorted. Therefore, the word is always the two letters in order, so the word is the same regardless of the original pair. Therefore, the word is determined by the two letters, but sorted. Therefore, the word is the two letters in order. So, the word is the same for any pair of letters, but the pair is unordered. Therefore, the word is the two letters in order. So, the word is the concatenation of the two letters in alphabetical order. Therefore, the word is uniquely determined by the two letters, but the pair is unordered. Therefore, the word is the same for any pair of letters, but the pair is just the two letters. So, the key is that the word is the two letters sorted, so the word is the same as the pair sorted. Therefore, the word is the two letters in order, so the word is the same as the pair sorted. Therefore, the word is the two letters in order, so the word is the same as the pair sorted. Therefore, the word is the two letters in alphabetical order, so the word is the same as the pair sorted…
>
> This trace illustrates that the model has a kind of risk-aversion, similar to the hardness based mechanism we discuss: the model repeatedly takes a cyclic action — re-explaining how pairs are sorted into words—instead of a progress-making action, such as identifying the key structural insights needed to solve the problem. For example, in this problem, a progress-making action would be to observe that for the last word to contain G, no word may start with a letter after G.

---

> ### Author Response · Authors · 2025-11-20
>
> **Further analysis of example traces:**
>
> To probe this further, we truncate the student’s trace at the first looping point (the first time it says ``Wait, so actually, the word is always the two letters in alphabetical order, so the word is the pair sorted.’’) and let the teacher model (QwQ-32B) complete from that prefix under greedy decoding. The teacher’s continuation does not loop and does reach progress-making actions such as those above.
>
> We also quantify how much probability mass student and teacher place on progress-making actions. Starting from the same looping prefix, we sample 50 continuations of length 2000 tokens each from both the student and the teacher at a high temperature. We then feed these continuations to GPT-5 and ask whether each contains any of the key insights (the progress making actions) needed to make progress on the problem. For the student, only 1/50 continuations contained such an insight, whereas for the teacher 34/50 continuations did. Thus, in line with our hardness-of-learning mechanism, the student puts significantly less mass on the progress-making action.
>
> Across other traces we examined, this risk-averse behavior, where the student repeatedly revisits a cyclic action instead of committing to a progress-making one, was the most common pattern. In one trace, we observed a different pattern closer to our “temporally correlated errors” mechanism: the model makes a mistake in the proof, builds on it until it reaches a contradiction, then takes a cyclic action by going back and redoing the proof, but repeats the same mistake again, reaching the same contradiction, and repeats these series of steps several times.
>
> We will add these example traces and their discussion to the final version of the paper and hope they help ground the proposed mechanisms further.
>
> > Connection with Ryan Cotterell's group’s work and other reasoning paradigms.
>
> We believe the reviewer is referring to recent work from Cotterell’s group on the computational power of language models with chain-of-thought reasoning  (e.g., [1]), which compares the expressivity of such models to probabilistic Turing machines and finite-state devices. Our star-graph setup is also a finite-state stochastic process, but our focus is not on characterizing the computational power of LMs; instead, we use this setting to isolate how learning errors and decoding dynamics give rise to looping behavior. Thus, to the best of our understanding, the connection between the two is not very deep.
>
> For other reasoning paradigms such as MCTS-style reasoning, our expectation is that the specific looping problem we study should be much less prominent, since cyclic operations like backtracking are explicitly encoded in the search procedure and controlled by the algorithm. In contrast, the looping we study arises from the model’s learned distribution combined with low-temperature sampling in free-form chain-of-thought generation.
>
> [1] https://arxiv.org/pdf/2406.14197
>
> > What is the difference between repetition in standard text generation, and reasoning generation? What characteristics of reasoning make the repetition more (or fewer)?
>
> This is a great question. We hypothesize that looping is significantly more prevalent in reasoning models because of the abundance of natural cyclic actions in reasoning (e.g., restating past facts, re-checking computations, or self-evaluation). This is partly supported by past work [2], which shows that the fraction of repetitions in a model’s output correlates with how prevalent those repetitions are in the training data. A similar dependence shows up in our controlled study: in the hardness-based mechanism, if we decrease the probability of the cyclic action (going back to the source), then at some point it becomes low enough that the hard progress-making action, even with its probability diffused over alternatives, still wins out, and looping disappears.
>
> [2] https://arxiv.org/abs/2310.10226
>
> > Why no evaluation on closed-source models?
>
> We did not evaluate closed-source reasoning models for several practical reasons. Most of them do not reveal the full reasoning traces, many do not allow us to control the sampling temperature, and model sizes and training details are often unknown. In the absence of these controls, it is difficult to perform a meaningful and comparable study of looping behavior across models.

---

> > ### Comment · Reviewer_Lw8Q · 2025-11-21
> > **Adequate responses. Increasing from 6 -> 8.**
> >
> > Hi authors,
> >
> > Thank you for your responses in great detail and depth!
> >
> > I read them and I think the AIME 2025 example is particularly helpful for linking your setup and real world problem. So I decide to raise my score.
> >
> > I am now wishing to champion this paper into acceptance. I believe this paper starts from the first principle and has built a very interesting stimuli. It is a novel effort and will benefit the field.
> >
> > Best regards,

---

### Official Review · Reviewer_ZsM5 · 2025-10-30

**Soundness:** 3
**Presentation:** 3
**Contribution:** 3
**Rating:** 6
**Confidence:** 3

**Summary:**

This paper investigates the phenomenon of looping behavior in reasoning LLMs, where models repeatedly generate similar or identical text segments during inference. The authors analyze several open-source reasoning models and observe that looping occurs more frequently at lower decoding temperatures, in smaller-capacity models, and in distilled student models compared to their teachers. The authors attribute these behaviors to two underlying mechanisms: Hardness of Learning, where models overfit easy but unproductive reasoning paths when true progress steps are difficult to learn, and Temporally Correlated Errors, where small prediction biases compound across time, causing self-reinforcing repetition under deterministic decoding. These two mechanisms are demonstrated through a simplified graph-traversal setup(star graph).

**Strengths:**

- **Novel and Relevant Problem**: The paper tackles looping, a highly observed failure mode in reasoning LLMs,  and tries to identify and understand this issue.
- **Empirical Breadth Across Open Models**: The authors study looping across a variety of open-source LLM families, sizes, and training paradigms.
- **Star-Graph Demonstration as an Effective Illustration**: While the star-graph environment itself is not novel, the paper uses it effectively as a minimal and controlled demonstration tool to isolate the proposed mechanisms, which are themselves meaningful contributions.
- **Rich Analyses and Interpretive Insights**: The paper offers a wide range of analyses and observations that go beyond a single factor such as temperature. It provides a more holistic understanding of looping behavior across different conditions
- **Clarity**: The paper is clear. The writing, structure, and figures make the argument easy to follow.

**Weaknesses:**

- **Limited Dataset Diversity**: The experiments use the AIME math problems (~30 items). While each problem is complex and multi-step, broader reasoning domains (e.g., logic, science, code) would confirm its generality. The paper can benefit from including additional datasets to demonstrate that the observed looping mechanisms generalize beyond a single domain.
- **Looping detection**: Even though the n-gram repetition for k times is practical, it does detect semantic loops(the model repeats the same idea using different tokens).
- **Limited Generalization**: While the star-graph setup effectively isolates looping mechanisms, it remains simplistic and may not fully capture the richness of reasoning trajectories in real-world LLM tasks.

**Questions:**

- How sensitive are your looping measurements to the specific definition of “loop” (e.g., n-gram size, k repetition threshold)? Could small changes in these detection parameters significantly alter reported looping frequencies, especially for longer CoTs?
- Given that the Phi model deviates from the general looping–temperature trends, do you expect similar family-specific differences to arise in other reasoning models trained under distinct architectures or data regimes?
- Do you think your findings would extend to MoE reasoning models?

---

> ### Author Response · Authors · 2025-11-20
>
> Thank you for your positive comments, for appreciating the empirical breadth of our open-model experiments, and for highlighting the usefulness of the star-graph setup for understanding looping behavior. Your main concerns were about relying on AIME as the primary dataset and the robustness of our looping definition. In the response below, we add additional results on GPQA, present further experiments studying the sensitivity of our looping metric, and discuss your other questions around semantic looping and the role of data and architecture. We hope these additions help address your concerns and we look forward to hearing from you.
>
> # Experiments beyond AIME
>
> We do additional evaluations with GPQA-Diamond (a challenging multiple-choice question set in biology, chemistry, and physics, authored by PhD-level experts) on a subset of the models. We see the same qualitative trends here (smaller models looping more; student looping more than teacher). We report some representative results for zero temperature below and will also include them with more details in the final version of the paper.
>
> | Model Name | Looping Fraction at T=0 |
> | :--- | :--- |
> | Qwen 2.5 Distill-1.5B | 0.77 |
> | Qwen 2.5 Distill-7B | 0.41 |
> | Qwen 2.5 Distill-32B | 0.26 |
> | OpenThinker 3 1.5B | 0.74 |
> | OpenThinker 3 7B | 0.23 |
> | OpenThinker 3 QwQ-32B | 0.07 |
>
> Since reasoning models produce a huge number of tokens, their evaluation is expensive (for example, the gpu cost of running our current evaluation experiments once is ~10k USD, and with several iterations and debugging, it becomes significantly more). Therefore, we had to make a decision on which datasets and models to focus on. We decided to consider an extensive set of models, while for datasets, we restricted ourselves to AIME 2024 and 2025, which are arguably the most used datasets for evaluating reasoning models. Nevertheless, we agree that it would be valuable to understand looping dynamics in other domains and we hope future work will build on our analysis.
>
>
> # How sensitive are the conclusions to the definition of looping?
>
> Our main results mark a response to be looping if there an n-gram repeats at least k times in the model response, where we set $n=30$ and $k=20$. However, our conclusions (e.g., that smaller models in the same family loop more, that the student can loop significantly more than teacher, etc.) are not too sensitive to these exact values of $n$ and $k$.
> For example, in Figure 10 of Appendix, we include plots with values of $k$ increased to 60. Further, we add further experiments varying $n$ to $20$ and $40$, and again see that the conclusions remain the same. We report the numbers with different values of $n$ below, and will include them in the final version of the paper.
>
> **Looping Fraction Results (n=20, k=30)**
> | Model Name | Looping Fraction |
> | :--- | :--- |
> | Qwen 2.5 Distill-1.5B | 0.77 |
> | Qwen 2.5 Distill-7B | 0.50 |
> | Qwen 2.5 Distill-32B | 0.30 |
> | OpenThinker 3 1.5B | 0.27 |
> | OpenThinker 3 7B | 0.11 |
> | OpenThinker 3 QwQ-32B | 0.00 |
> | Llama 3.1 Distill-8B | 0.52 |
> | Phi-4 reasoning | 0.30 |
> | Phi-4 reasoning-plus | 0.31 |
>
>
> **Looping Fraction Results (n=40, k=30)**
>
> | Model Name | Looping Fraction |
> | :--- | :--- |
> | Qwen 2.5 Distill-1.5B | 0.76 |
> | Qwen 2.5 Distill-7B | 0.48 |
> | Qwen 2.5 Distill-32B | 0.27 |
> | OpenThinker 3 1.5B | 0.25 |
> | OpenThinker 3 7B | 0.10 |
> | OpenThinker 3 QwQ-32B | 0.00 |
> | Llama 3.1 Distill-8B | 0.49 |
> | Phi-4 reasoning | 0.27 |
> | Phi-4 reasoning-plus | 0.29 |
>
>
> **Looping Fraction Results (n=20, k=60)**
>
> | Model Name | Looping Fraction |
> | :--- | :--- |
> | Qwen 2.5 Distill-1.5B | 0.75 |
> | Qwen 2.5 Distill-7B | 0.44 |
> | Qwen 2.5 Distill-32B | 0.23 |
> | OpenThinker 3 1.5B | 0.24 |
> | OpenThinker 3 7B | 0.10 |
> | OpenThinker 3 QwQ-32B | 0.00 |
> | Llama 3.1 Distill-8B | 0.41 |
> | Phi-4 reasoning | 0.27 |
> | Phi-4 reasoning-plus | 0.29 |
>
>
> **Looping Fraction Results (n=40, k=60)**
>
> | Model Name | Looping Fraction |
> | :--- | :--- |
> | Qwen 2.5 Distill-1.5B | 0.72 |
> | Qwen 2.5 Distill-7B | 0.40 |
> | Qwen 2.5 Distill-32B | 0.18 |
> | OpenThinker 3 1.5B | 0.23 |
> | OpenThinker 3 7B | 0.10 |
> | OpenThinker 3 QwQ-32B | 0.00 |
> | Llama 3.1 Distill-8B | 0.37 |
> | Phi-4 reasoning | 0.27 |
> | Phi-4 reasoning-plus | 0.28 |
>
>
>
> # Detecting semantic looping
>
> Since semantic looping is more subjective, quantifying it is challenging. Therefore, in our open-model experiments we focus on quantifying syntactic loops. That said, we agree that analyzing semantic looping is worthwhile, and we hope our work serves as a jumping-off point for future work in this direction.

---

> > ### Author Response · Authors · 2025-11-20
> >
> > > Do you expect similar family-specific differences to arise in other reasoning models trained under distinct architectures or data regimes?
> >
> > We think our work has only scratched the surface, and more work is needed to fully understand this question. Our expectation is that the data distribution can have a significant impact on looping behavior. For example, our controlled-setting experiments show that looping depends on the presence of hard actions and on the probability of cyclic actions. Since different data-generation regimes can alter these factors, it is certainly possible that changing the data distribution changes looping behavior.
> > In the discussion section, we also outline potential ways of mitigating the underlying causes of looping (hard actions) by making the data distribution easier for the student to learn. Overall, we believe there is substantial room for follow-up work to explore how different training data and regimes affect looping.
> >
> >
> > > Do you think your findings would extend to MoE reasoning models?
> >
> > One nice property of MoE reasoning models is that they offer additional capacity without significantly increasing inference cost. Since our findings indicate that increased model capacity helps reduce looping, it is plausible that MoE models may loop less than their non-MoE counterparts when we control for the number of active parameters at inference time. When we instead control for total model capacity, we do not have a strong reason to expect MoEs to perform better. In general, it is a fascinating direction to understand how different architectural components (including MoE) affect looping and its underlying causes.
> >
> > > While the star-graph setup effectively isolates looping mechanisms, it remains simplistic and may not fully capture the richness of reasoning trajectories in real-world LLM tasks.
> >
> > We agree, and this is precisely our intention with the controlled experiments. Given the complexity of modern LLMs, it is often difficult to make progress on understanding any phenomenon directly. Controlled setups such as our graph reasoning task help by isolating the phenomenon of interest while leaving aside much of the other complexity. Further, we performed an additional analysis of example traces (see our response to Reviewer Lw8Q for an example) and found that our observations in the controlled setup do map to real traces. We will add these example traces to the final version of the paper.

---

### Official Review · Reviewer_yggL · 2025-10-30

**Soundness:** 2
**Presentation:** 3
**Contribution:** 3
**Rating:** 4
**Confidence:** 4

**Summary:**

This paper investigates the phenomenon of looping, repetitive token generation often observed in reasoning models such as DeepSeek-R1 or Phi-4 Reasoning, particularly under greedy decoding and low temperatures. The authors conduct a large-scale empirical study showing that smaller and distilled reasoning models loop more frequently than their larger counterparts, and that temperature generally mitigates looping. To explain this, the paper proposes a graph-based formalization of reasoning as a random walk over nodes, demonstrating two mechanisms through which errors in learning can lead to loops: (1) hardness of learning, where a hard-to-learn “progress action” has its probability diffused across many alternatives while an easy cyclic action dominates; and (2) temporal correlation of errors, where small correlated deviations across time steps cause repetitive selections. The authors further discuss implications for model training, temperature tuning, and reasoning stability.

**Strengths:**

1. **Timely and valuable question**: Understanding why reasoning models loop is a meaningful and underexplored problem. The looping issue directly relates to the stationarity of model decoding dynamics and the stability of chain-of-thought reasoning.

2. **Interesting research direction**: Formalizing decoding as a graph traversal is conceptually insightful. It provides a potential framework to reason about “deadlock” loops in probabilistic generation processes.

3. **Comprehensive empirical evaluation**: The analysis across model families (Qwen, Phi-4, OpenThinker, LLaMA) and the controlled graph experiments are carefully conducted and informative.

4. **Clear empirical trend**: The paper convincingly shows that looping decreases with temperature and model capacity, reinforcing the empirical validity of the observation.

**Weaknesses:**

1. **Gap between graph formalization and language model decoding.**
The connection between the proposed graph reasoning setup and actual language model decoding remains vague. While the analogy to random walks is appealing, the paper does not rigorously justify how node transitions correspond to token-level or reasoning-step-level dynamics in real models. The mapping between “actions” in the graph and “reasoning paths” in language models needs clearer formal grounding. Incorporating perspectives such as finite-state automata (FSA) or Markov chain abstractions could make this connection more rigorous.

2. **Unclear or overly intuitive definitions.**
The key definitions (e.g., “hard actions”, “cyclic actions”, “progress-making actions”) are intuitive but lack formal precision. The notion of “reasoning” itself is used informally, and the theoretical treatment risks being too heuristic for a work positioned as a formal analysis. More careful mathematical framing and clearer assumptions would strengthen the paper.

3. **Proof of Proposition 1 is unclear.**
The derivation in Appendix C appears inconsistent. For example, in the cross-entropy expression, the second term should arguably be $-(1-p)\log{1-q_0}$ rather than $-(1-p)\frac{1}{n}\sum_{i=1}^{n} \log{q_i}$. If the authors intend to split the hard-action probability among n indistinguishable actions, the derivation should clearly justify each transformation step. Revisiting this proof and illustrating the simplification process would improve transparency.

4. **Conceptual overlap with prior works.**
The two looping mechanisms, learning hardness and correlated errors, partly restate well-known properties of autoregressive models (e.g., local maxima trapping, exposure bias). The novelty is thus limited unless the graph formalization offers a more rigorous bridge to these effects.

5. **Writing clarity.**
While the paper is well-organized, some sections (especially section 3 and 4) are dense with narrative explanations but light on formal statements. More concise mathematical exposition and illustrative examples would enhance readability.

**Questions:**

Could the authors formalize how the random-walk state transitions relate to token-level probability distributions in LLM decoding?

Would viewing the process as a finite-state or Markov model help in proving convergence/stationarity properties more rigorously?

Could the authors validate their theoretical claims with diagnostic traces from real LLM decoding (e.g., token probability trajectories during loops)?

---

> ### Author Response · Authors · 2025-11-20
>
> Thank you for carefully reading our paper. We are happy that you found the questions we study timely, the graph-based formalization conceptually insightful, and the empirical evaluation across model families informative. Your main concerns were around the clarity of our definitions, the mapping between the controlled setup and language model decoding, the proof of Proposition 1, and the extent of conceptual overlap with prior work. In the response below, we make the key definitions more precise, add example traces to further ground the mapping, clarify the proof of Proposition 1, and explain how our mechanisms relate to existing concepts. We hope these clarifications address your concerns and we look forward to your feedback.
>
>
> # Clarifying the definitions and the mapping
>
> The notions of actions, cyclic actions, and progress-making actions can be precisely stated in our graph reasoning task. There, an action is simply the choice of the next node to visit from the current node. A cyclic action is an action that takes the walk to an already visited node, and a progress-making action is an action that moves the walk closer to the goal node.
>
> The analogous notions for reasoning language models can be defined at the level of short spans of tokens (``chunks’’). A chunk is a span of tokens that implements a single logical step in the chain-of-thought (for example, a line or small paragraph). Conditioned on the prefix generated so far, a language model induces a distribution over possible next chunks; these are the actions in our abstraction. A cyclic action corresponds to a chunk that repeats something already done before (e.g., restating a previously stated fact), while a progress-making action corresponds to a chunk that moves closer to the final goal (e.g., taking the next step in the proof or narrowing down the candidate set).
>
> We agree that in language models these notions are inherently less precise, and this is exactly why we use the controlled graph reasoning task: it lets us give these concepts sharp definitions and analyze them cleanly, while still capturing the looping patterns we see in real models. Upon reading your review, we realized that perhaps we did not state these notions clearly enough even for the graph reasoning task. Thank you for pointing this out. In the final version, we will explicitly state the precise definitions of actions, cyclic actions, and progress-making actions in the graph setting as discussed above.
>
> **On the Markovian perspective.**
> Below, we further clarify how this mapping interacts with the proposed mechanism, and how the Markovian perspective fits in. Our hardness-of-learning mechanism can be broken down into two components.
>
> First, when a hard action co-occurs with an easy action, the probability of the hard action is diffused across many indistinguishable alternatives, so the easy action becomes relatively high probability. This effect is captured and formalized by Proposition 1 and does not rely on any Markovian assumption. In our graph reasoning task, this plays out at the root node: there is a single progress-making action among many indistinguishable alternatives, and a separate easy action (going back to source node) whose probability is not similarly diffused. The same idea applies to language models where actions are chunks of tokens and we discuss this at the end of Proposition 1.
>
> Second, when the easy action is cyclic, its relatively higher probability gives rise to loops. This is particularly easy to see in a Markovian view: in our graph reasoning task, if the student (similar to the teacher)  induces a Markov chain over the nodes and the cyclic action (taking the walk back to the source) dominates at the root, then under greedy decoding the walk will tend to bounce between the source and the root.
>
> However, note that this Markovian view is not fully accurate. Transformer models are not Markovian since their next action can depend on the full history rather than only the last node. Nonetheless, our experiments with transformer models on the graph tasks show that, even without an exact Markov assumption on the student, the same kind of looping behavior emerges. Thus, while the Markovian student assumption is not fully accurate, it helps in understanding the mechanism, and our experiments demonstrate that the mechanism also manifests without such an assumption. We thank the reviewer for suggesting that this perspective may make the illustration clearer, and in the final version we will make this view explicit so that the mechanism is easier to follow.

---

> ### Author Response · Authors · 2025-11-20
>
> # Example traces
> To further understand the mapping between our mechanisms and actual reasoning models, we examined some looping reasoning traces from the OpenThinker-3 1.5B model. We discuss one example trace below.
>
> The model is solving the following AIME 2025 problem:
>
> > The twelve letters $A$, $B$, $C$, $D$, $E$, $F$, $G$, $H$, $I$, $J$, $K$, and $L$ are randomly grouped into six pairs of letters. The two letters in each pair are placed next to each other in alphabetical order to form six two-letter words, and then those six words are listed alphabetically. For example, a possible result is $AB,CJ,DG,EK,FL,HI$. The probability that the last word listed contains $G$ is $\\frac{m}{n}$, where $m$ and $n$ are relatively prime positive integers. Find $m + n$.
>
> First, the trace restates the problem and correctly counts the total number of pairings:
>
> >First, let me make sure I understand the problem correctly. We have the letters A, B, C, D, E, F, G, H, I, J, K, and L. These are to be randomly paired up into six pairs. Each pair is then arranged in alphabetical order to form a two-letter word. Then, all these six words are listed alphabetically. The question is asking for the probability that the last word in this list contains the letter G.
>
> >…
>
> > Let me start with the total number of ways to pair the letters. Since the letters are being paired into unordered pairs, the number of ways to partition 12 letters into 6 pairs is given by (12)! divided by (2^6 * 6!)....
>
> Next, it tries to reason about how the two-letter words are formed and loops (semantically) for a bit about alphabetical ordering of pairs:
>
> > Each pair is formed by taking two letters and putting them in alphabetical order. So, for any pair, say, (X, Y), if X comes before Y in the alphabet, then the word is XY; otherwise, it's YX. Wait, but actually, the problem says "the two letters in each pair are placed next to each other in alphabetical order to form six two-letter words". So, regardless of the order in the pair, they are arranged in alphabetical order. So, for any pair, the two letters are ordered such that the first is before the second in the alphabet. Therefore, each pair is effectively an ordered pair in the alphabetical order. So, for example, if the pair is (C, D), the word is CD. If the pair is (D, C), the word is DC, but since they are placed in alphabetical order, it's DC? Wait, no, actually, if the pair is (D, C), then since D comes after C, they would be placed as CD? Wait, no, hold on. Wait, the problem says "the two letters in each pair are placed next to each other in alphabetical order". So, regardless of the original order of the pair, they are arranged in alphabetical order. So, for any pair, the two letters are sorted, so the first letter is the earlier one, the second is the later one. So, for example, if the pair is (C, D), the word is CD. If the pair is (D, C), the word is DC. Wait, but actually, the pair is just two letters, so the pair is unordered, but when forming the word, they are arranged in alphabetical order…
>
> Later, the model comes out of this semantic loop and outlines a high-level counting strategy. After that, it again falls into a loop, repeatedly re-deriving the same fact about sorted pairs, and never gets out of it:
>
> > Therefore, to compute this probability, we can think of all possible pairings, and count how many pairings result in the maximum word in the set containing G, then divide by the total number of pairings.
>
> > ...
>
> > First, let me note that G is the 7th letter. So, the letters before G are A, B, C, D, E, F, and after G are H, I, J, K, L. So, G is in the middle. So, the pairs that include G can be either (G, something before G) or (something before G, G). But when sorted, the pair with G will be either (G, something before G) or (something before G, G) but sorted, so it will be (something before G, G). Wait, no. Wait, if the pair is (G, H), then the word is GH. If the pair is (H, G), the word is HG, but since they are sorted, it's GH. Wait, so actually, the word is always the two letters in alphabetical order, so the word is the pair sorted. Therefore, the word is always the two letters in order, so the word is the same regardless of the original pair. Therefore, the word is determined by the two letters, but sorted. Therefore, the word is the two letters in order… Therefore, the word is uniquely determined by the two letters, but the pair is unordered. Therefore, the word is the same for any pair of letters, but the pair is just the two letters. So, the key is that the word is the two letters sorted, so the word is the same as the pair sorted. Therefore, the word is the two letters in order, so the word is the same as the pair sorted. Therefore, the word is the two letters in order, so the word is the same as the pair sorted. Therefore, the word is the two letters…

---

> ### Author Response · Authors · 2025-11-20
>
> The above trace illustrates that the model has a kind of risk-aversion, similar to the hardness based mechanism we discuss: the model repeatedly takes a cyclic action --- re-explaining how pairs are sorted into words --- instead of a progress-making action, such as identifying the key structural insights needed to solve the problem. For example, in the problem, a progress-making action would be to observe that for the last word to contain G, no word may start with a letter after G.
>
>
> **Further analysis of example traces:**
> To probe this further, we truncate the student’s trace at the first looping point (the first time it says ``Wait, so actually, the word is always the two letters in alphabetical order, so the word is the pair sorted.’’) and let the teacher model (QwQ-32B) complete from that prefix under greedy decoding. The teacher’s continuation does not loop and does reach progress-making actions such as those above.
>
> We also quantify how much probability mass student and teacher place on progress-making actions. Starting from the same looping prefix, we sample 50 continuations of length 2000 tokens each from both the student and the teacher at a high temperature. We then feed these continuations to GPT-5 and ask whether each contains any of the key insights (the progress making actions) needed to make progress on the problem. For the student, only 1/50 continuations contained such an insight, whereas for the teacher 34/50 continuations did. Thus, in line with our hardness-of-learning mechanism, the student puts significantly less mass on the progress-making action.
>
> Across other traces we examined, this risk-averse behavior, where the student repeatedly revisits a cyclic action instead of committing to a progress-making one, was the most common pattern. In one trace, we observed a different pattern closer to our “temporally correlated errors” mechanism: the model makes a mistake in the proof, builds on it until it reaches a contradiction, then takes a cyclic action by going back and redoing the proof, but repeats the same mistake again, reaching the same contradiction, and repeats these series of steps several times.
>
> We will add these example traces and their discussion to the final version of the paper and hope they help ground the proposed mechanisms further.
>
> # Proof of proposition 1
> The proof was indeed skipping over some important steps, so we re-wrote it. Thank you for helping us improve the transparency and presentation of our paper.
>
> For completeness, we include the new proof below and will include it in the final version of the paper.
>
> Let $q_1, \dots, q_n$ be the probabilities that the model assigns to the $n$ actions, and $q_0$ be the probability of the easy action. The Cross-Entropy loss for a context $c_i \in C_i$ with correct action $a_i$ is
> $$
>         \ell_i = -p \log(q_0) - (1 - p) \log(q_i).
> $$
>     Since all $n$ contexts are equi-probable, the average Cross Entropy over the dataset will be
> $$
>   \begin{align*}
>          \ell &= \frac{1}{n} \sum_{i=1}^n \ell_i\\\\
>             &= -p \log(q_0) - (1 - p) \frac{1}{n} \sum_{i=1}^n \log(q_i).
>     \end{align*}
> $$
>     Note that $\sum_{i=0}^n q_i = 1$ and $q_i \geq 0$ for all $i$, by definition. We can further assume that $q_i > 0$, otherwise the loss would be infinite. This is a convex minimization problem under linear constraints, so to find the minimizer it suffices to look at the points satisfying the KKT conditions. In particular,
> $$
>  \begin{align*}
>         &\frac{p}{q_0} = \mu \implies q_0 = \frac{1}{\mu} p\\\\
>         &\frac{1 - p}{n} \frac{1}{q_i} = \mu \implies q_i = \frac{1}{\mu} (1 - p) / n \ \text{ for $i = 1, \ldots, n$},
>     \end{align*}
> $$
>     for some $\mu \in \mathbb{R}$. Using the equality constraint, we conclude that $\mu = 1$ and $q_0 = p$, $q_i = (1 - p) / n$ for $i = 1, \ldots, n$ is the unique minimizer of the problem, as desired.
>
>
>
> # Conceptual overlap with prior works
> While the proposed mechanisms may sound similar to known concepts, to the best of our knowledge these similarities are only superficial. For example, exposure bias typically refers to the train–test mismatch induced by teacher forcing: at training time the model is conditioned on ground-truth prefixes, whereas at inference time it conditions on its own (possibly erroneous or different) generations, which can push it into out-of-distribution territory. The temporally-correlated-errors mechanism we study, while it may sound superficially similar, is quite different. In our setting, looping arises even when the student generates initial tokens that are well aligned with the teacher distribution; the key issue is that its estimation errors at repeated decision points are correlated over time, not that it is operating on out-of-distribution prefixes.

---

### Official Review · Reviewer_8CqX · 2025-10-31

**Soundness:** 2
**Presentation:** 2
**Contribution:** 2
**Rating:** 2
**Confidence:** 4

**Summary:**

This paper investigates why looping behavior appears in chain-of-thought (CoT) reasoning models. The authors find that looping primarily arises hard and biased predictions.

**Strengths:**

Extensive experiments are done across both large-scale reasoning LLMs and small models trained from scratch.

**Weaknesses:**

The findings are not particularly interesting. They confirm intuitive explanations (e.g., low temperature amplifies biases, worse models tends to loop more) rather than offering new theoretical insight or actionable mitigation strategies.

**Questions:**

What decoding method is used in the experiments? It seems to be greedy decoding, but temperature should not affect greedy decoding, since the argmax operation is invariant to temperature scaling.

---

> ### Author Response · Authors · 2025-11-20
>
> Thank you for your comments. We discuss them below.
> > ``Findings confirm intuitive explanations rather than offering new theoretical insight’’
>
> While interestingness is subjective, the take that this paper just confirms intuitive explanations rather than offering new insights feels unfair. To the best of our knowledge, many fundamental questions around the phenomenon of looping are still poorly understood. For instance, what are the fundamental causes of looping? Is randomness/temperature fixing the underlying cause, or is it merely a stopgap? Our paper makes progress on these questions, shedding light on mechanisms behind looping and the role of randomness.
>
> > ``What decoding method is used in the experiments? It seems to be greedy decoding, but temperature should not affect greedy decoding, since the argmax operation is invariant to temperature scaling.’’
>
> In all our experiments, temperature = 0 corresponds to greedy decoding, and for higher temperature, we do sampling. So there is no argmax operation involved at non-zero temperatures.

---

### Official Review · Reviewer_vtW9 · 2025-11-02

**Soundness:** 2
**Presentation:** 2
**Contribution:** 2
**Rating:** 4
**Confidence:** 4

**Summary:**

This paper investigates why reasoning language models (LLMs) exhibit persistent "looping", repetitive output generation, especially at low sampling temperature or under greedy decoding. Through empirical evaluations on a large suite of open-source reasoning models on math benchmarks (AIME problems), the authors document that looping is severe in smaller or distilled models, lessens with model scale, and is largely absent in their teacher or instruction-tuned counterparts. A key finding is that temperature reduces looping but may introduce its own accuracy trade-offs. The authors then introduce controlled graph-reasoning experiments to demonstrate two underlying error mechanisms: (1) "hardness of learning," where progress actions are hard to distinguish, resulting in probability being diffused across alternatives, and (2) "temporally correlated estimation errors," which become amplified into loops under certain sampling regimes. The implications for future model training and inference practices are discussed.

**Strengths:**

1. The paper contains a thorough and systematic empirical study of looping behavior in a wide array of modern reasoning-focused LLMs, including multiple sizes, instruction/distilled variants, and reinforcement learning (RL)-tuned models, using a challenging math dataset.
2. Proposition 1 formalizes the hardness of learning scenario with a precise and correct description; the proof in Appendix C is sound and maps cleanly to LLM training as described in Section 3. The connection between indistinguishability, softmax allocation, and looping is well justified.
3. The graph-reasoning tasks in Section 3.1 and Section 4 demonstrate the identified error mechanisms in highly controlled settings, making the analysis and implications compelling. Figure 4 and Figure 5 (as well as their supporting metrics) structurally isolate how action ambiguity or error correlation leads to looping and are valuable for the transparency and reproducibility of the arguments.

**Weaknesses:**

1. The core observation, looping/repetition at low temperatures, has been well documented for LLMs since Holtzman et al. (2020) and subsequent works, though the controlled graph experiments and direct focus on "reasoning" models add new clarity. The main conceptual contributions lie more in consolidating, empiricizing, and framing recent issues than in presenting an entirely new mechanism or solution.
2. Though the focus on AIME questions is appropriate, the generalizability of findings to domains beyond math reasoning is left largely unexplored. All main looping/accuracy/temperature analyses (Figures 1–2) are on this single dataset; it would be valuable to know whether looping dynamics differ in language, code, or multimodal settings (see e.g., how behavioral features in FlowVQA/other logic-heavy tasks compare).
3. Contemporary approaches such as contrastive decoding, unlikelihood training, entropy penalization, or dynamic decoding strategies are mentioned but not directly compared as baselines in the experiments (Section 2 and Appendix B). Readers are left to infer whether these approaches can alleviate looping on reasoning tasks beyond what temperature sampling achieves.
4. The bulk of mitigation discussion focuses on adjusting sampling temperature, but could more structured mechanisms, e.g., entropy regularization, adaptive n-gram avoidance, or neural decoding algorithms, also be relevant? The absence of empirical ablation or theoretical remarks on these means the prescription feels somewhat narrow.

Holtzman, Ari, et al. "The curious case of neural text degeneration." arXiv preprint arXiv:1904.09751 (2019).

**Questions:**

See weakness

---

> ### Author Response · Authors · 2025-11-20
>
> Thank you for taking the time to read our paper. We are glad you found the empirical study and the controlled star-graph analysis compelling. Your main concerns were around the conceptual contributions, the focus on AIME as the primary dataset, and our emphasis on temperature as the main mitigation. In the response below, we (i) clarify our contributions, (ii) add additional evaluations on GPQA, and (iii) explain why we focus on temperature, including an additional experiment with repetition penalty, and discuss the outstanding questions that remain beyond temperature-based mitigation. We hope these clarifications address your concerns and we look forward to hearing your feedback.
>
> # On the motivation and contributions of the paper
>
> Looping in language models at low temperatures has indeed been a known issue for several years. With recent instruct models, the issue seemed to be going away (as we also show in our empirical study), until reasoning models appeared, where looping again became very prevalent. While high temperature or randomness seems to fix the issue in practice, many basic questions remain poorly understood (for reasoning models and even more generally). For instance, what are the fundamental causes of looping? Is randomness fixing the underlying cause, or is it merely a stopgap? The main goal of our paper is to make progress on these questions. In particular, the mechanisms we propose are one of our central contributions and, to the best of our knowledge, are novel.
>
> For clarity, we list all our contributions below:
>
> **Extensive empirical study of looping in reasoning models.** Our first contribution is an extensive empirical study measuring looping in a variety of reasoning models (e.g., OpenThinker, R1-distilled-Qwen, R1-distilled-Llama, Phi-4 reasoning). This leads to many interesting observations: reasoning models loop even when their instruct variants barely loop; harder problems induce more looping; smaller models in the same family tend to loop more, etc. Perhaps the most telling observation is that for models trained with distillation, student models show significant looping even when the teacher barely loops. This strongly suggests that imperfect learning or errors in learning are a key cause of looping (if the student had perfectly learned the teacher distribution, it would not loop significantly more).
>
> **Understanding mechanisms behind looping.** Our second contribution is to demonstrate several mechanisms by which errors in learning cause looping. For this, we propose a controlled graph reasoning setup that may be of independent interest. Among the mechanisms, we show that hard-to-learn points in reasoning traces can induce looping, and that transformers also have an inductive bias toward looping even in the absence of such hardness.
>
> **Understanding the role of randomness.** Finally, our third contribution is to clarify the role of randomness. We show that while high temperature or randomness can decrease looping, it does not fix the underlying hardness-induced errors. As a result, even at high temperatures, the student model produces significantly longer traces than the teacher. We close with a discussion of training-time fixes (better datasets and architectures) that can attack these underlying causes more directly.
>
> To summarize, while the existence of looping is well known, our main contribution is to (i) identify and empirically characterize how it manifests in modern reasoning models, (ii) propose concrete mechanisms that explain why it arises, and (iii) analyze the role of randomness/temperature in mitigating—but not fundamentally resolving—these mechanisms.

---

> ### Author Response · Authors · 2025-11-20
>
> # Beyond math reasoning
>
> We do additional evaluations with GPQA-Diamond (a challenging multiple-choice question set in biology, chemistry, and physics, authored by PhD-level experts) on a subset of the models. We see the same qualitative trends here (smaller models looping more; student looping more than teacher). We report some representative results for zero temperature below and will also include them with more details in the final version of the paper.
>
> | Model Name | Looping Fraction at T=0 |
> | :--- | :--- |
> | Qwen 2.5 Distill-1.5B | 0.77 |
> | Qwen 2.5 Distill-7B | 0.41 |
> | Qwen 2.5 Distill-32B | 0.26 |
> | OpenThinker 3 1.5B | 0.74 |
> | OpenThinker 3 7B | 0.23 |
> | OpenThinker 3 QwQ-32B | 0.07 |
>
> Since reasoning models produce a huge number of tokens, their evaluation is expensive (for example, the gpu cost of running our current evaluation experiments once is ~10k USD, and with several iterations and debugging, it becomes significantly more). Therefore, we had to make a decision on which datasets and models to focus on. We decided to consider an extensive set of models, while for datasets, we restricted ourselves to AIME 2024 and 2025, which are arguably the most used datasets for evaluating reasoning models. Nevertheless, we agree that it would be valuable to understand looping dynamics in other domains and we hope future work will build on our analysis.
>
>
> # Beyond temperature as a mitigation
>
> We focus on temperature because it is the de facto method used by essentially all open reasoning model providers to control looping, yet its effects are still poorly understood. Beyond being the standard practical control knob, it is also scientifically appealing because it touches on a fundamental question: is randomness a necessary resource for good reasoning models?
>
> Along with temperature, some reasoning model providers use a repetition penalty that downweights tokens that have occurred previously. Among the models we test, only the OpenThinker family uses it. To test its effectiveness independently of temperature, we ran an additional experiment where we used a repetition penalty with greedy decoding for the OpenThinker-3 1.5B model. We observed that this frequently led to gibberish generations (including lots of Chinese characters) and significantly more looping compared to using no repetition penalty. Thus, it does not appear very effective on its own.
>
> Finally, note that temperature is in fact reasonably effective in decreasing looping. What our controlled experiments point to is that while high temperature significantly reduces looping, it doesn’t solve the fundamental cause: hardness induced errors in learning. Because of this, while the model doesn’t loop much at high temperatures, it still produces significantly longer responses than its teacher. Thus the outstanding question is which measures can help solve this underlying hardness? Other known mitigation strategies such as repetition penalty, unlikelihood training, contrastive decoding etc. are not aimed at solving these hardness induced errors, and are thus unlikely to be effective. Among the measures that can help are better training datasets that explicitly address these hard points, or better architectures, as we discuss in the final section of the paper.
>
> From reading your review, we realized that this distinction between "what temperature actually fixes" and "what remains unsolved" may not have been sufficiently clear in our current draft.  Thank you for pointing it out. In the final version, we will add the additional experiment with repetition penalty and expand the discussion of outstanding challenges and their relation to existing strategies,  so that readers are not left guessing about what our analysis suggests for these other methods.

---

### Meta-Review · Area_Chair_cEm7 · 2026-01-13

**Summary:**

This paper investigates important phenomenon in modern "reasoning" language models (e.g., DeepSeek-R1, Phi-4): the tendency to enter repetitive loops, particularly at low temperatures. The authors provide a strong empirical characterization of this behavior, noting that higher capacity models loop less and it is more prevalent in distilled "student" models, suggesting imperfect learning as a root cause.

Thanks for the authors for active participation during the rebuttal period (e.g. adding datasets (GPQA) and correcting some misunderstanding). But, in my opinion, paper ultimately falls short of the novelty and depth required for acceptance. The core phenomenon of looping at low temperatures is a fairly well-known observation of AR language models, extensively documented since Holtzman et al. (2020). Applying this known observation in the reasoning setup confirms intuition but does not generate significantly new insights. Furthermore,, the paper focuses on the problem without offering a concrete, novel mitigation strategy beyond suggesting well-known strategy of higher temperature, In my opinion, the paper is not ready for publication. I recommend rejection.

**Reviewer Concerns:**

Reviewer 8CqX: Found the findings "not particularly interesting" and merely confirming intuition.

Reviewer vtW9 had concerns about the novelyt of the work and narrow empirical analysis of the paper (which is still a concern in my opinion).

Reviewer yggL raised concerns about the presentation of the work and Proposition 1. The reviewer mentioned that the definitions of "cyclic actions" were somewhat vague.

Reviewer ZsM5 raised concern about limited dataset diversity (only AIME) and the sensitivity of the n-gram looping metric. In response, the authors provided sensitivity analysis and added GPQA and sufficiently addressed this concern.

**Reviewer Scores:**

Reviewer 8CqX: The rebuttal additional data does not fundamentally change thie concern about novelty. the low score remains.

Reviewer vtW9: The addition of addition experiments addresses the "limited scope" weakness, which would likely be enough to flip a borderline reviewer.

Reviewer yggL: The Reviewer would probably keep the same score.

Reviewer ZsM5: Would probably keep their slightly high score.

Overall the paper will still be borderline reject.

---

### Decision · Program_Chairs · 2026-01-26

Reject